# Physical Trajectory Inference Attack and Defense in Decentralized POI Recommendation

## ABSTRACT

As an indispensable personalized service within Location-Based Social Networks (LBSNs), the Point-of-Interest (POI) recommendation aims to assist individuals in discovering attractive and engaging places. However, the accurate recommendation capability relies on the powerful server collecting a vast amount of users' historical check-in data, posing significant risks of privacy breaches. Although several collaborative learning (CL) frameworks for POI recommendation enhance recommendation resilience and allow users to keep personal data on-device, they still share personal knowledge to improve recommendation performance, thus leaving vulnerabilities for potential attackers. Given this, we design a new Physical Trajectory Inference Attack (PITA) to expose users' historical trajectories. Specifically, for each user, we identify the set of interacted POIs by analyzing the aggregated information from the target POIs and their correlated POIs. We evaluate the effectiveness of PITA on two real-world datasets across two types of decentralized CL frameworks for POI recommendation. Empirical results demonstrate that PITA poses a significant threat to users' historical trajectories. Furthermore, Local Differential Privacy (LDP), the traditional privacy-preserving method for CL frameworks, has also been proven ineffective against PITA. In light of this, we propose a novel defense mechanism (AGD) against PITA based on an adversarial game to eliminate sensitive POIs and their information in correlated POIs. After conducting intensive experiments, AGD has been proven precise and practical, with minimal impact on recommendation performance.

## 1 INTRODUCTION

The next Point-of-Interest (POI) recommendation recently continues to bloom due to the widespread application of Location-based Social Networks (LBSNs) such as Weeplace and Foursquare. Aiming to understand users' behavioral patterns and predict their preferences for the next movement, the next POI recommendation has diverse applications like urban planning, mobility prediction, and location-based advertising [13]. Benefiting from constantly improving computing resources to collect and process massive training data, current attention-based neural networks have excelled in delivering quality recommendations [20, 32, 38]. Unfortunately, sustaining such a powerful central server comes at a financially and environmentally high cost. In addition, the service timeliness is extremely precarious as it depends on the internet quality for uploading requests and downloading results. More notably, due to the growing emphasis on privacy and the special sensitivity of real-world trajectories, the POI recommender faces more difficulties in centrally obtaining users' personal check-in histories, thus hindering the recommendation quality.

Hence, the decentralized collaborative learning (CL) paradigms emerge for POI recommendation, aspiring to address the shortcomings of centralized frameworks. Specifically, a precise yet resource-efficient recommendation model is deployed on end devices for instantaneous inference regardless of the network quality. In addition, models are all locally trained and further improved by exchanging knowledge with other users. Therefore, users' data is kept on the device, significantly reducing the potential for privacy breaches. As a widely recognized approach in CL, federated learning (FL) based POI recommenders (e.g., [10]) employ a cloud server to collect and aggregate all locally trained models, subsequently redistributing the aggregated model to all users. Although the FL paradigm has impressive performance regarding flexibility and generalization capability, it still heavily relies on the central server for model aggregation and redistribution. More importantly, the paradigm of all users sharing a standard model inevitably tends to favor popular POIs, leading to sub-optimal performance.

To achieve a higher degree of personalization, users within the same group of the decentralized POI recommender [17] are allowed to collaborate in an end-to-end manner. This diagram effectively reduces reliance on the central server, demonstrated by the server only being in charge of model initialization and neighbor identification. However, even with intra-group communication, models are also optimized by sharing parameters/gradients, causing high communication costs between devices. Moreover, this requires that all models must be structurally equivalent, weakening the applicability of the decentralized diagram, as in the real world, mobile devices possess various hardware configurations and the assumption will restrict the overall performance to the capability of the worst device. As the remedy, Long et al. [18] further proposed a decentralized POI recommender that supports collaborative learning between heterogeneous models, allowing users to customize the model configurations. Here, locally trained models are enhanced by the knowledge distillation mechanism, where users only need to share their soft decisions on a public reference dataset rather than models/gradients, enabling efficient communication.

With the attention towards CL continuing to increase, recent research has begun investigating the level of safety it can provide. Among all the aforementioned CL methods, FL has the highest privacy risk as the server fully controls the learning process and has access to all models. Consequently, there are various security and privacy attacks against FL frameworks (e.g., Poisoning Attacks [8, 24] and Inference Attacks [25]). There has been a slight alleviation of the privacy risks in decentralized CL after eliminating the central server. However, this framework still requires exchanging personal knowledge (e.g., model parameters or soft decisions) within groups, leaving vulnerabilities for privacy attacks [1, 22].

As a representative and valuable threat, inference attacks aiming to infer user information, have been widely investigated in a number of CL-based recommendation tasks [8, 19, 24, 36]. However, they are all dedicated to traditional e-commerce recommendations instead of POI recommendations, while the latter is more noteworthy considering the service's location-sensitive nature. Once users' physical trajectories are exposed to attackers, their property and even their personal safety will be compromised. In detail, knowing users' visited places allows attackers to track their movements and

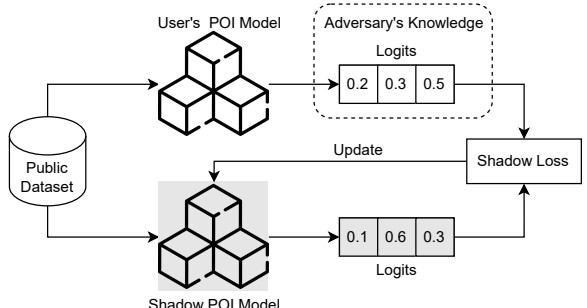

**Figure 1: Build the Shadow POI Model for Knowledge-Distillation-based CL.**

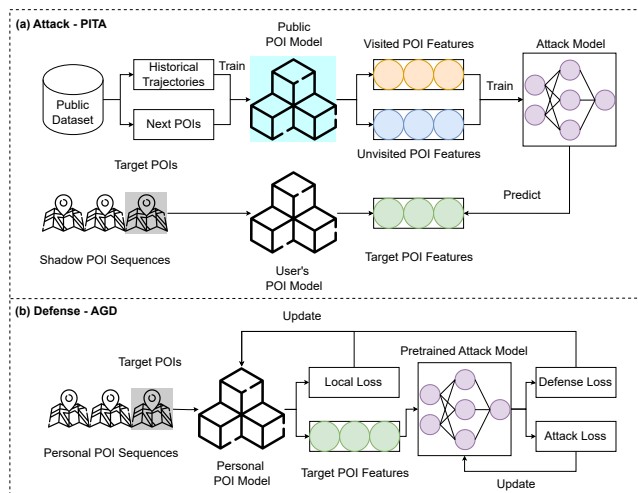

**Figure 2: The overview of PITA (part a) and AGD (part b). Please note that defense loss and attack loss are contrary numbers.**

daily routines, which can be exploited for various malicious purposes such as harassment and revealing sensitive information (e.g., medical facilities visited). Besides, identifying shared places among individuals can provide insights into users' social connections and networks, potentially exposing social ties that users may wish to keep private. More notably, the information of POIs can be not only represented by their own embeddings but also inferred from the embeddings of related POIs, which is ignored by existing inference attacks on e-commerce CL recommendations.

On this basis, we propose a novel attack, namely Physical Trajectory Inference Attack (PTIA) to reveal users' interacted POIs in decentralized POI recommendations, followed by an effective defender. Decentralized POI recommenders can be classified into two types based on different collaboration learning methods including (1) **Model-Sharing-based CL** sharing model parameters, and (2) **Knowledge-Distillation-based CL** sharing soft decisions on the reference dataset. To make our attack compatible in both cases, as shown in Figure 1, the Knowledge-Distillation-based CL is converted to the first type after building the shadow models to simulate users' true models by minimizing their response discrepancies on the reference dataset.

Given the user model, as demonstrated in Figure 2(a), whether the target user has visited a specific POI can be inferred from its final feature generated by this model. However, this feature is not only related to the POI itself but also significantly influenced by related POIs in the same sequence, where the sequence is kept on-device, being unavailable to the attacker. As a replacement, the shadow target sequence is obtained by combining anonymous sequences containing the target POI, or assigning POIs to anonymous category sequences with geographical restrictions when the former is infeasible. The aggregated sequence can moderately reveal the real preferences of the target user since two users who have visited the same POI may have similar preferences, and thus, mutual information exists in their historical trajectories. It is worth noting that we employ a Multi-Layer Perceptron (MLP) as the attack model.

To speed up the attack algorithm, instead of repeating the above process to all POIs, we first identify visited regions by quantifying and comparing the distances from the embeddings of all POIs within each region to their corresponding initial embeddings, and then detect interacted POIs in those regions. After conducting extensive experiments on two datasets, the high inference accuracy

has validated the effectiveness of PITA. Even employing Local Differential Privacy (LDP), which gains much attention by providing valid privacy protection of federated recommendations [27], the inference accuracy is almost unaffected unless there is a significant sacrifice of recommendation precision.

Hence, we further propose a defense mechanism against PITA based on an adversarial game (AGD), shown in Figure 2(b). Specifically, we innovatively integrate PITA into the training process as the attack model. Whenever the selected POI is involved in training, the probability of being visited which is printed by the adversary must be the same level as unrelated POIs. Meanwhile, the attack model is further improved by the visited features returned by the POI model. In this way, not only the selected POI but also its implicit information within related POIs will be eliminated. The experiment results show that the proposed defense mechanism can nearly paralyze PTIA for selected POIs with negligible sacrifice in recommendation quality. In conclusion, our contributions can be summarized as follows:

- To the best of our knowledge, we are the first to explore privacy concerns related to users' real mobility trajectories in decentralized collaborative learning-based POI recommendations. Meanwhile, we propose a novel physical trajectory inference attack (PITA), aimed at identifying users' interacted POIs in this scenario.
- As the defense mechanism against PITA, we design a novel adversarial game (AGD) to eliminate sensitive POIs and their implicit information within related POIs from the personal POI recommenders before they are shared for collaborative learning.
- After conducting extensive experiments with two real-world datasets, we have demonstrated the effectiveness of our attack (PITA) and defense (AGD) approaches in decentralized POI recommendation.

## 2 PRELIMINARIES

In this section, we first introduce important notations used in this paper and then formulate our major tasks. Let $\mathcal{U}$, $\mathcal{P}$, and $C$ denote the sets of users $u$, POIs $p$ and categories $c$, respectively. Each POI $p \in \mathcal{P}$ is associated with a category tag (e.g., entertainment or restaurant) $c_p \in C$ and coordinates $(lon_p, lat_p)$.

**Definition 1: Check-in Sequence**. A check-in activity of a user indicates a user $u \in \mathcal{U}$ has visited POI $p \in \mathcal{P}$ at timestamp $t$. By sorting a user's check-ins chronologically, a check-in sequence contains $M_i$ consecutive POIs visited by a user $u_i$, denoted by $\mathcal{X}(u_i) = \{p_1, p_2, ..., p_{M_i}\}$.

**Definition 2: Category Sequence**. A category sequence substitutes all POIs in the check-in sequence $\mathcal{X}(u_i)$ with their associated category tags, denoted by $\mathcal{X}^c(u_i) = \{c_{p_1}, c_{p_2}, ..., c_{p_{M_i}}\}$.

**Definition 3: Reference Dataset**. The reference dataset $\mathcal{D} = \{\mathcal{X}_z\}_{z=1}^{Z}$ contains $Z$ anonymous check-in sequences covering all POIs $\mathcal{P}$.

**Definition 4: Region**. A region $r$ is essentially a geographical segment that can provide additional information about the POIs within it. Without any assumptions on predefined city districts/suburbs, we obtain a set of regions $\mathcal{R}$ by applying $k$-means clustering [21] on all POIs' coordinates in our paper.

**Task 1: Physical Trajectory Inference Attack**. As mentioned above, we focus on two types of decentralized CL frameworks for POI recommendations including model-sharing-based CL and knowledge-distillation-based CL. In both cases, the adversary's goal is uniformly to infer the set of interacted POIs for all clients. Formally, given a target user $u_i$, and the knowledge $\mathcal{K}_i$ obtained by the adversary regarding $u_i$, for each POI $p_m \in \mathcal{P}$, the physical trajectory inference attack $\mathcal{A}$ can be defined as:

$$\mathcal{A} : u_i, \mathcal{K}_i, p_m \rightarrow (0, 1), \quad (1)$$

where **1** indicates $u_i$ has visited $p_m$, while **0** is the opposite. However, the attack knowledge obtained by the adversary is different for the two cases:

- **Model-Sharing-based CL**: In this case, users' personal models will be sent to other users, and thus, we can assume the attacker can get the model parameters of the target user $u_i$, denoted as:

$$\mathcal{K}_i^{ms} = \Theta_i. \quad (2)$$

- **Knowledge-Distillation-based CL**: Instead of the model itself, the attacker here can only get the soft decisions on the reference dataset from the target user $u_i$, which can be defined as:

$$\mathcal{K}_i^{kd} = \{\Theta_i(\mathcal{X}), \mathcal{X} \in \mathcal{D}\}. \quad (3)$$

Beyond that, we can assume multiple anonymous sequences containing the target POI are publicly available to the adversary for both cases. This is reasonable because some LBSs (e.g., Weeplace and Foursquare) have exposed desensitized datasets. This assumption can be further relaxed by assigning POIs to category sequences with geographical restrictions.

**Task 2: Protect Sensitive POIs Against PITA**. The defense mechanism is performed on-device to avoid the potential trajectory inference attack on sensitive POIs from other clients after they receive parameters or soft decisions. Intuitively, given the set of sensitive POIs $\mathcal{H}_i$ for $u_i$, our task is to erase them and their information hidden within the relevant POIs from the locally trained model $\Theta_i$ before it or its based soft decisions are shared for collaborative learning. As such, the adversary cannot detect whether $u_i$ has visited those POIs in any case.

## 3 METHOD

This section first introduces the local objective function that facilitates the model optimization on each user's device, and two types of decentralized CL frameworks for POI recommendations, followed by the details of the physical trajectory inference attack against those two frameworks, as well as the corresponding defense mechanism.

### 3.1 Local Objective Function

The main objective of this work is to launch the trajectory inference attack and defender at the personalized POI recommenders, which are locally trained with users' private check-in sequences under the guide of the local objective function:

$$L_{loc}(u_i) = l\left(\Theta_i\left(\mathcal{X}(u_i)\right), \mathcal{Y}(u_i)\right), \quad (4)$$

where $\Theta_i\left(\mathcal{X}(u_i)\right)$ is the prediction made by the recommender $\Theta_i(\cdot)$ given $\mathcal{X}(u_i)$. In the POI recommendation setting, the predictions are made successively on historical POI sequences $\{p_1\}, \{p_1, p_2\}, ..., \{p_1, p_2, ..., p_{M_i-1}\}$, and $\mathcal{Y}(u_i) = \{p_2, p_3, ..., p_{M_i}\}$ is the set of corresponding ground truth POIs. $l$ is the loss function (i.e., cross-entropy in our case) to quantify the prediction error. It is worth noting that the proposed attack and defense can be applied to most deep neural networks for POI recommendations.

### 3.2 Collaborative Learning Protocols

To alleviate data sparsity caused by training personal models solely on the device end, those models will be further optimized by collaborative learning with others. In this work, we perform the privacy analysis on two types of decentralized CL-based POI recommenders. (1) **DCLR** [17]. Models in this framework are shared within groups in an end-to-end manner. (2) **MAC** [18]. Instead of sharing models like DCLR, similar users communicate with each other by extracting knowledge from soft decisions on the reference datasets. In summary, to improve recommendation performance, users need to share either models or prediction results on reference datasets, leaving attack vulnerabilities.

### 3.3 Physical Trajectory Inference Attack

Given the model $\Theta_i$ or soft decisions $\{\Theta_i(\mathcal{X}), \mathcal{X} \in \mathcal{D}\}$, the adversary aims to infer the interacted POIs of the user $u_i$. To achieve this, for a specific POI $p_m$, we focus on exploring its explicit information and implicit information within related POIs (i.e., POIs in the same sequence). Intuitively, we propose to distinguish between the interacted POI and irrelevant POIs by comparing their final features after applying the model $\Theta_i$ on carefully designed sequences, which requires frequent access to the model. Obviously, it is infeasible to directly deploy this attack on MAC as the attacker can only get soft decisions rather than the model from the target user. Consequently, for MAC, we first utilize shared soft decisions to build the shadow

model. Specifically, we let the shadow model increasingly approach the user model by minimizing their disagreement of prediction results on $\mathcal{D}$ via the following:

$$L_{shadow} = \sum_{X \in \mathcal{D}} \left\| \Theta_i'(X) - \Theta_i(X) \right\|_2^2, \tag{5}$$

where $\Theta_i'$ represents the shadow recommender of the local recommender $\Theta_i$ possessed by $u_i$. Now, the attacker has the model for model-sharing-based CL (DCLR) or the shadow model for knowledge-distillation-based CL (MAC), putting them in a similar situation for the rest of the attack. For convenience, they will be uniformly denoted as $\Theta_i$.

With the user model $\Theta_i$ ready, the new challenge is getting the specific sequence containing the target POI while revealing the user's preference, which is unavailable in CL frameworks. Alternatively, the anonymous sequence containing the target POI can be regarded as the shadow sequence. This is because, If two users have visited the same POIs, their historical trajectories might contain implicit information about each other. Intuitively, the more shadow sequences there are, the higher the possibility of getting close to the target user. Thus, for the target user $u_i$ and POI $p_m$, we prepare multiple shadow sequences, denoted by $\mathcal{S}(u_i, p_m)$. In case we cannot collect enough shadow sequences, we can build them by randomly assigning POIs to the anonymous category sequence $X^c$ with the restrictions of containing the target POI and the distance between any two consecutive POIs being less than 5km, where the category sequence can be obtained more easily from desensitized datasets.

Given the user model $\Theta_i$ and the set of shadow sequences $\mathcal{S}(u_i, p_m)$, a Multi-Layer Perceptron (MLP) is further adopted as the attack model $\mathcal{A}$ to predict whether $u_i$ has visited $p_m$. The input is the average feature of applying $\Theta_i$ to all sequences in $\mathcal{S}(u_i, p_m)$, denoted as:

$$Input(u_i, p_m) = \frac{1}{V} \sum_{X \in \mathcal{S}(u_i, p_m)} \Theta_i(X), \tag{6}$$

where $V$ is the number of shadow sequences in $\mathcal{S}(u_i, p_m)$. Besides, the output is a two-dimensional vector where the first $(\alpha_n^0)$ indicates the probability of having been to $p_m$ while the second $(\alpha_n^1)$ is the probability of the opposite. The attack model $\mathcal{A}$ is trained with the binary cross-entropy loss:

$$L_{pita} = -\sum_{n=1}^{N} (y_n log \alpha_n^1 + (1 - y_n) log \alpha_n^0), \tag{7}$$

where $N$ is the number of training sample and $y_n$ is the ground truth label. Those training samples come from a public POI model $\Theta_{public}$ trained with anonymous check-in sequences from publicly available datasets. For each sequence $X = \{p_1, p_2, ..., p_M\}$ that is involved in the training process of $\Theta_{public}$, we can obatain one positive training sample of $\mathcal{A}$ where the input is the feature of $p_M$ after applying $\Theta_{public}$ to $X$. We also sample an equivalent number of sequences that are not included in the training process of $\Theta_{public}$, and those sequences can be utilized to produce negative training samples of $\mathcal{A}$ in the same way above.

Then, we can get visited POIs for $u_i$ by applying $\mathcal{A}$ to all POIs. Although this approach is straightforward, it becomes inefficient with the vast number of POIs, requiring a substantial amount of computational resources. To tackle this problem, we design a novel strategy to detect $u_i$' visited regions $\mathcal{R}_i$, which can effectively reduce the scope of attack implementation. In decentralized CL frameworks, all models are initialized with the same distribution. Once POIs are involved in the training process, their parameters will naturally deviate from this distribution. Thus, regions, where POI representations are far away from the initial distribution, can be identified as visited regions. Formally, for each region $r \in \mathcal{R}$, we first adopt the Kullback-Leibler (KL) divergence [9] to quantify the differences between the embeddings of all POIs $e_r$ within each region and the initial distribution $e_r'$:

$$d_r = KL\left(e_r' \| e_r\right). \tag{8}$$

Then, inspired by the elbow method [23], we sort all regions by their distances to the initial distribution in descending, and mark regions as visited until the distances start to converge. Given visited regions $\mathcal{R}_i$, we apply $\mathcal{A}$ to POIs in those regions to obtain interacted POIs for $u_i$.

## 3.4 Defender: An Adversarial Game to Protect Sensitive POIs Against PITA

After conducting extensive experiments in Section 4.4, we have demonstrated the effectiveness of the proposed attack, which poses a significant threat to users' privacy for two types of decentralized CL frameworks, proving the necessity of the defense mechanism. A widely accepted strategy is Local Differential Privacy (LDP) [27] which adds a specific level of noise to personal models before sharing them with others. Unfortunately, as indicated by the experiment results shown in Table 3, evenly disturbing all model parameters is invalid since the low noise level does not impact the attack accuracy of PITA. On the other hand, a sufficiently high noise level will significantly sacrifice the recommendation precision, rendering the POI recommender meaningless.

In this situation, hiding a few sensitive POIs rather than all visited POIs seems a better way to reach the balance between recommendation performance and privacy protection. As the key of the POI recommender, the trained POI embeddings are highly likely to disclose user privacy, and hence, a simple approach to hide sensitive POIs is replacing their trained embeddings with initial ones. However, this cannot completely obliterate traces of sensitive POIs as their information not only exists with their own embeddings but is also revealed by embeddings of related POIs. In addition, the final recommendation accuracy is inevitably affected by this abrupt information loss.

On this basis, we design a novel adversarial game (AGD) to erase sensitive POIs $\mathcal{H}_i$ and their implicit information in related POIs from the locally trained model $\Theta_i$. In this game, we have innovatively integrated PTIA $\mathcal{A}_i$ into the training process as an adversary of the POI recommender. Whenever sensitive POIs are involved in the training of the POI recommender, the probability of those POIs being visited, which the adversary predicts, must be the same level as unrelated POIs. In light of this, the defense loss is

**Algorithm 1** The workflow of local model training with the proposed defense mechanism. All processes are implemented on the user's device.

1: **for** $u_i \in \mathcal{U}$ **do in parallel**
2:     Initialize $\Theta_i$ and $\mathcal{A}_i$;
    /*Pretraining the attacker with shared samples*/
3:     **repeat**
4:         Take a gradient w.r.t $L_{pita}$ to update $\mathcal{A}_i$;
5:     **until** convergence
6:     **repeat**
    /*Training the POI recommender while erasing sensitive POIs*/
7:         Take a gradient w.r.t $L_{loc} + \mu L_{def}$ to update $\Theta_i$ ;
    /*Fine-tuning the attacker with samples produced by $\Theta_i$*/
8:         Take a gradient w.r.t $L_{pita}$ to update $\mathcal{A}_i$;
9:     **until** convergence
10:    Share $\Theta_i$ or soft decisions generated by $\Theta_i$ for collaborative learning;
11: **end for**

defined as:

$$L_{def} = \frac{1}{G} \sum_{p \in \mathcal{H}_i} \left| \mathcal{A}_i(p) - \frac{1}{O} \sum_{o=1}^{O} \mathcal{A}_i(p_o) \right|, \qquad (9)$$

where $G$ is the number sensitive POIs in $\mathcal{H}_i$. For each POI $p \in \mathcal{H}_i$, we randomly sample $O = 5$ unrelated POIs. Meanwhile, the adversary is also improved with samples produced by the recommender. In this manner, both the POI recommender and the attacker can grow stronger through continuous adversarial encounters. Consequently, we not only eliminate those sensitive POIs but also remove the implicit information embedded within related POIs. It is also worth noting that, in real life, users can customize sensitive POIs according to their needs. Differently, $G$ sensitive POIs in this work will be randomly selected based on their global access frequency, meaning that the less frequently accessed POI will have a higher probability. The rationale is, users tend to prefer hiding private, niche POIs rather than popular ones.

The workflow of local model training with the proposed defense mechanism is presented in Algorithm 1. For each user, we first initialize the POI recommender $\Theta_i$ and the adversary $\mathcal{A}_i$, as well as pretraining the attacker (lines 2-5). The pretraining samples are obtained from a shadow POI recommender which is trained on the desensitized datasets which is also described in Section 3.3. This process can be accomplished personally on the device or delegated to the server. Then, for each epoch, we first train the personal recommender $\Theta_i$ with the synergic loss $L_{loc} + \mu L_{def}$ for achieving better recommendation and eliminating sensitive POIs and their implicit information in related POIs where $\mu$ controls the extend of the attacker's participation in training (line 7). After that, the attacker $\mathcal{A}_i$ is further improved with the samples produced by $\Theta_i$ (line 8). Finally, the privacy-preserving model or its based soft decisions are shared for collaborative learning (line 10).

## 4   EXPERIMENTS

In this section, we conduct comprehensive experiments to evaluate the performance of the physical trajectory inference attack (PITA),

and the effectiveness of the corresponding defense mechanism based on an adversarial game (AGD).

**Table 1: Dataset statistics.**

|  | Foursquare | Weeplace |
|---|---|---|
| #users | 7,507 | 4,560 |
| #POIs | 80,962 | 44,194 |
| #categories | 436 | 625 |
| #check-ins | 1,214,631 | 923,600 |
| #check-ins per user | 161.80 | 202.54 |

### 4.1   Datasets and Evaluation Protocols

We adopt two real-world datasets, Foursquare [31] and Weeplace [16], and both of them consist of users' check-in histories in the cities of New York, Los Angeles and Chicago. Inspired by [2, 13], we remove users and POIs with less than 10 interactions for better data quality. The statistics of the two datasets are summarized in Table 2. Among these, 15% of the check-in sequences will be utilized as prior knowledge for the attacker while an additional 15% will serve as the reference dataset for MAC. Each of them is selected randomly satisfying the constraint of including all POIs. As for the remaining sequences, we adopt the leave-one-out strategy [26] for the evaluation of recommendation performance. That is, for each sequence, the last POI is for testing, the second last is for validation, and all the others are for training. Furthermore, we set the maximum sequence length to 200. Following [17], we compare each ground truth with the 200 nearest unvisited POIs as potential candidates for ranking, with the expectation that the ground truth will be ranked at the top. Then, we employ Hit Ratio at Rank k (HR@K) [28] to evaluate the recommendation performance, which quantifies the proportion of the ground truth that appears in the top-k recommendation list.

To assess the effectiveness of PTIA, for each user, we first predict her visited regions. Then, we need to predict whether the user has been to each POI of those regions. Since this is a classification task, we adopt the commonly used F1 score [36, 37] as the metric to evaluate the attack performance. In addition, for any incorrectly predicted region, its corresponding F1 score is 0. The situation varies when evaluating the performance of the defense mechanism. As our goal is to hide sensitive POIs, we only measure the F1 score of those specific POIs.

### 4.2   Baselines

As mentioned above, the proposed attack and defender are applied to two decentralized CL frameworks including DCLR and MAC. Besides, we prepare three attacker baselines and two defender baselines. Here, the adversary's knowledge is unified as the personal model for each user.

**Attacker Baselines**:

- **Random Attack**: The set of interacted POIs is randomly selected from $\mathcal{P}$ with equal probability, while the set size is obtained by averaging true set sizes across all users.
- **K-means Attack**: Since all POI embeddings are available to the attacker, we can directly adopt the k-means [21] algorithm to divide POIs into two clusters while the cluster with

lower SSE (i.e., the sum of squared errors) will be selected as the set of visited POIs. The rationale is, for a user, positive POIs exhibit higher similarity to each other compared to diverse negative items, owing to the coherence principle of personal interests.

- **Interaction-level Membership Inference Attack (IMIA)** [36]: The set of interacted POIs is obtained by comparing uploaded POI embeddings and trained POI embeddings with public sequences.

**Defender Baselines**:

- **Local Differential Privacy (LDP)**: Being a widely adopted method to protect users' personal data, LDP has been incorporated into many federated recommender systems [27]. The core is to add noise before the model $\Theta_i$ is shared for CL:

$$\Theta_i \leftarrow \Theta_i + \mathcal{N}(0, \lambda^2 I), \tag{10}$$

where $\mathcal{N}$ refers to the normal distribution while $\lambda$ decide the level of noise.

- **Embedding Reset (ER)**: A simple approach to hide sensitive POIs is to replace their trained embeddings with the initial embeddings before the model $\Theta_i$ is shared for CL.

### 4.3 Experimental Setting

First of all, we adopt STAN [20] as the base POI recommender for advanced accuracy. Then, as explained in Section 2, we divide each city into 5 regions with k-means clustering [21]. Besides, the attack model is an MLP with 3 hidden layers having 64, 32, and 8 units, respectively. Since MAC supports heterogeneous structures, we randomly assign the latent dimension $d \in \{8, 16, 32, 64, 128\}$ to users and each one makes up 20%. For fairness, DCLR is evaluated with the above dimensions separately and final results are averaged. For hyperparameters, we set $V$ to 5, and $\mu$ to 0.6, while the impacts of the two hyperparameters will be further discussed in Section 4.7. Apart from this, we set the learning rate to 0.002, the dropout ratio to 0.2, the batch size to 16, and the maximum training epoch to 50. It's worth mentioning that all experimental results are obtained by averaging multiple trials.

### 4.4 Attack Performance of PTIA

**Table 2: The performance (F1 scores) of attackers.**

| Model | Attack | Foursquare | Weeplace |
|-------|--------|-----------|----------|
| | Random | 0.0354 | 0.0751 |
| DCLR | K-means | 0.1695 | 0.1634 |
| | IMIA | 0.3201 | 0.3172 |
| | PITA | 0.6103 | 0.6389 |
| MAC | K-means | 0.1741 | 0.1867 |
| | IMIA | 0.2926 | 0.3185 |
| | PITA | 0.5321 | 0.5413 |

Table 2 summarizes four attackers' performances on two decentralized POI recommenders and two datasets, where we have the following observations. In POI recommendation, decentralized CL frameworks without deliberate protection pose a significant risk of

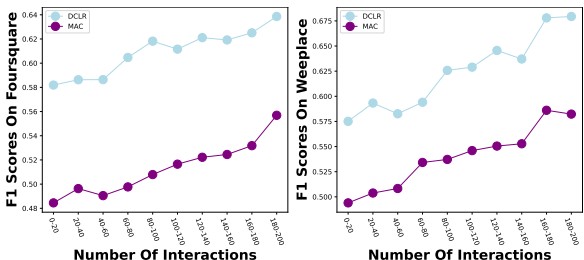

**Figure 3: PITA performance for users with different numbers of interactions.**

exposing users' historical trajectories, which is demonstrated by the fact that both K-means attack and PITA have better performance than Random attack. While the K-means attack has been shown to have some effect, its low attack accuracy lacks practical significance in real-life scenarios. In addition, IMIA falls significantly short of PITA in terms of attack accuracy. This is because IMIA solely focuses on variations in the parameters of target POIs while PITA combines explicit information from the target POI and implicit information within related POIs, thus proving the effectiveness of PITA.

Among the two decentralized CL frameworks, as expected, the proposed PITA has a worse performance on MAC. This can be explained by the fact that, in MAC, we can only infer users' models based on their response to the reference data, failing to fully restore users' preferences. Even though PITA underperforms on MAC compared to DCLR, the historical trajectories of MAC's users are still glaringly exposed under PITA. Besides, PITA has higher attack accuracy on Weeplace than Foursquare. We believe the reason is that users in Weeplace have more interactions with POIs, which contain interconnected information. To further prove this view, we cluster users into 10 groups and record the attack accuracy of PITA (F1 Score). The results are shown in Figure 3 where we can observe that users with more interactions have a higher risk of leakaging historical trajectories.

### 4.5 Defense Performance of LDP Against PITA

As mentioned above, LDP is widely applied to various CL frameworks due to its effectiveness in safeguarding user privacy. Hence, we conduct extensive experiments to explore whether LDP can serve as an effective defense mechanism against PITA. Specifically, table 3 shows the attack performance of PITA against LDP under different noise levels $\lambda \in \{0, 0.001, 0.01, 0.1\}$, where $\lambda = 0$ means that LDP is not implemented. The results indicate that when the noise level is low ($\lambda = 0.001$), not only does it fail to protect user privacy, but it also has a noticeable impact on recommendation accuracy in a negative manner. As the noise level increases to a significant extent ($\lambda = 0.1$), even though the attack performance of PITA significantly declines, the recommendation performance becomes alarmingly damaged. This is attributed to the fact that PITA is conducted by analyzing the information from target POIs and their correlated POIs. As a result, unless the noise level becomes substantial enough to make the recommender impractical, LDP remains ineffective against PITA.

**Table 3: The performance of Local Differential Privacy (LDP) against PITA.**

| Model | Dataset | Noise Level | | | | | | | |
|-------|---------|-------------|---|---|---|---|---|---|---|
| | | $\lambda=0.0$ | | $\lambda=0.001$ | | $\lambda=0.01$ | | $\lambda=0.1$ | |
| | | F1 | HR@10 | F1 | HR@10 | F1 | HR@10 | F1 | HR@10 |
| DCLR | Foursquare | 0.6103 | 0.4297 | 0.5988 | 0.3849 | 0.2768 | 0.2912 | 0.2001 | 0.2424 |
| | Weeplace | 0.6389 | 0.4664 | 0.6153 | 0.4478 | 0.4694 | 0.4237 | 0.3789 | 0.3557 |
| MAC | Foursquare | 0.5321 | 0.4319 | 0.5059 | 0.4216 | 0.2976 | 0.4187 | 0.1914 | 0.3156 |
| | Weeplace | 0.5413 | 0.4843 | 0.4791 | 0.4621 | 0.3242 | 0.4341 | 0.2813 | 0.3926 |

## 4.6 Defense Performance of AGD Against PITA

Since LDP fails to protect user privacy against PITA, we propose a novel adversarial game-based defense mechanism (AGD) to mitigate the exposure of sensitive POIs. Concurrently, we employ ER as a baseline, where we utilize initial embeddings to replace trained embeddings for sensitive POIS before knowledge sharing. Here, we explore the impact of these two defense approaches against PITA across varying numbers of sensitive POIs $G \in \{0, 1, 5, 10, 20\}$ where $G = 0$ means no defender is implemented. Table 4 and Table 5 record the results of ER and AGD respectively.

To begin with, for ER, the F1 score decreases as $G$ increases. This is because, when $G$ is small, replacing trained embeddings of sensitive POIs fails to eliminate the implicit information concealed within POIs correlated to sensitive POIs. When $G$ is large, some of those correlated POIs are also erased, preventing PITA from accurately inferring target POIs, ultimately leading to a lower F1 score. In contrast, the F1 score under AGD remains consistently low, regardless of the changes in $G$. This proves that AGD outperforms ER in precisely and effectively protecting sensitive POIs. More importantly, ER significantly impairs recommendation accuracy after abruptly erasing POIs for all values of $G$. When $G \geq 10$, the POI recommender becomes essentially non-functional. Conversely, AGD has a negligible impact on recommendation accuracy when $G \leq 10$. Even with a relatively large $G$ ($G \geq 10$), the POI recommender retains a certain level of functionality. To conclude, AGD has been proven as an effective and cost-saving defense mechanism.

## 4.7 Parameter Sensitivity

In this section, we further demonstrate the effect of two hyperparameters. First, we evaluate the impact of the number of shadow sequences $V \in \{1, 3, 5, 7, 9\}$ on the attack accuracy of PITA. Subsequently, we investigate how the weight $\mu \in \{0.2, 0.4, 0.6, 0.8, 1, 1.5\}$ affects the attack accuracy of PITA and its impact on recommendation accuracy, where $\mu$ controls the ratio of AGD involved in the training process. Please note that $G$ is set to 10 in this section. The results are shown in Figure 4.

**Impact of $V$.** From the attacker's perspective, the attack accuracy of PITA benefits from a higher value of $V$ for both decentralized frameworks, which proves the fact that a larger quantity of shadow sequences enhances the chance of approaching the target user. However, as $V$ reaches a certain level, the improvement tends to stop, and thus, it is essential to maintain $V$ within a reasonable range since higher levels of $V$ imply increased expenses.

**Impact of $\mu$.** From the user's perspective, we aim for lower attack accuracy and higher recommendation accuracy. For both decentralized frameworks, as AGD becomes more involved in the training process (higher $\mu$), the attack accuracy first decreases and then stabilizes at a lower level. Meanwhile, recommendation accuracy stabilizes at a higher level initially and then declines. Hence, we set $\mu = 0.6$ as the balance point, which can achieve a low attack accuracy and a high recommendation accuracy.

## 5 RELATED WORK

This section reviews recent literature on related areas including centralized models for POI recommendation, collaborative learning frameworks for POI recommendation, and attacks against collaborative learning frameworks in POI recommendation.

## 5.1 Next POI Recommendation

POI recommendation systems play an important role in helping people discover appealing and relevant locations. Early models utilized matrix factorization [15] and Markov Chains [6, 39] to capture correlations among users, POIs, and contextual features, while more recent approaches have employed recurrent neural networks (RNNs) to effectively capture spatiotemporal dependencies within sequences of POIs [4, 5, 12, 33, 35]. Additionally, innovative strategies like SGRec [14] constructed graph-augmented POI sequences, enhancing collaborative signals and achieving accuracy over RNN-based models. Attentive neural networks [3, 20, 32, 34] have also been integrated, employing self-attention layers to capture the relative spatiotemporal information of all check-in activities along the sequence. It's worth noting that the main contributions of this work lie in the security and privacy analysis of collaborative learning frameworks for POI recommendations, along with the proposed defensive measures, where most of the aforementioned models can be utilized as the base model. To obtain enhanced recommendation performance, we have selected STAN [20] as the base model for this work.

## 5.2 Collaborative Learning Frameworks for POI Recommendation

Collaborative learning frameworks have demonstrated their efficacy in overcoming the limitations of cloud-based learning for POI recommendations including the high cost of resources, weak resilience, and privacy issues. Specifically, Guo et al. [10] introduced a federated learning approach for POI recommendations, enabling edge servers to collect and aggregate locally trained models before distributing the aggregated model to all users. However, these approaches still exhibit a notable reliance on cloud servers. Subsequently, Jing et al. [17] designed a semi-decentralized learning paradigm with device collaboration. This paradigm empowers user devices to gather and merge knowledge from two distinct

Table 4: The performance of ER against PITA.

| Model | Dataset | Number of Sensitive POIs | | | | | | | | | |
| --- | --- | --- | --- | --- | --- | --- | --- | --- | --- | --- | --- |
| | | G = 0 | | G = 1 | | G = 5 | | G = 10 | | G = 20 | |
| | | F1 | HR@10 | F1 | HR@10 | F1 | HR@10 | F1 | HR@10 | F1 | HR@10 |
| DCLR | Foursquare | - | 0.4297 | 0.3213 | 0.4281 | 0.2938 | 0.3787 | 0.2427 | 0.3302 | 0.1849 | 0.3047 |
| | Weeplace | - | 0.4664 | 0.3521 | 0.4667 | 0.3924 | 0.4132 | 0.3012 | 0.3821 | 0.2145 | 0.2878 |
| MAC | Foursquare | - | 0.4319 | 0.3791 | 0.4219 | 0.3457 | 0.3785 | 0.2801 | 0.3483 | 0.1569 | 0.2712 |
| | Weeplace | - | 0.4843 | 0.4021 | 0.4683 | 0.3551 | 0.4072 | 0.2721 | 0.3721 | 0.2035 | 0.2983 |

Table 5: The performance of AGD against PITA.

| Model | Dataset | Number of Sensitive POIs | | | | | | | | | |
| --- | --- | --- | --- | --- | --- | --- | --- | --- | --- | --- | --- |
| | | G = 0 | | G = 1 | | G = 5 | | G = 10 | | G = 20 | |
| | | F1 | HR@10 | F1 | HR@10 | F1 | HR@10 | F1 | HR@10 | F1 | HR@10 |
| DCLR | Foursquare | - | 0.4297 | 0.1446 | 0.4327 | 0.1528 | 0.4185 | 0.1592 | 0.4083 | 0.1547 | 0.3523 |
| | Weeplace | - | 0.4664 | 0.1624 | 0.4598 | 0.1723 | 0.4628 | 0.1623 | 0.4313 | 0.1727 | 0.3782 |
| MAC | Foursquare | - | 0.4319 | 0.1433 | 0.4314 | 0.1524 | 0.4389 | 0.1485 | 0.4189 | 0.1578 | 0.3527 |
| | Weeplace | - | 0.4843 | 0.1541 | 0.4815 | 0.1624 | 0.4775 | 0.1727 | 0.4565 | 0.1562 | 0.4095 |

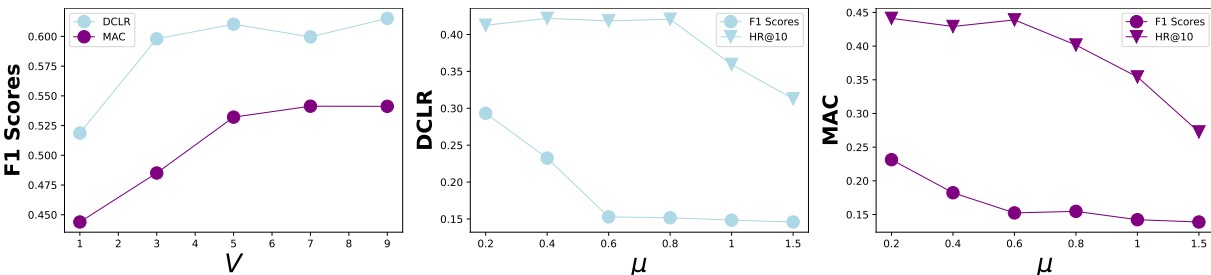

Figure 4: Hyperparameter sensitivity.

categories of neighboring devices. Nonetheless, the collaborative learning-based POI recommenders mentioned above operate under the assumption that all on-device models must adhere to an identical design, facilitating user-specific knowledge exchange through parameter/gradient aggregation. To address this issue, Jing et al. [18] further proposed a decentralized POI recommender, which grants users the ability to tailor model configurations according to their preferences. In this framework, locally trained models are further improved by the knowledge distillation mechanism. Instead of exchanging models/gradients, this mechanism only entails users sharing their soft decisions on a public reference dataset, thereby optimizing communication efficiency. Unfortunately, all the aforementioned methods require sharing user knowledge in various ways and extents, leaving vulnerabilities for potential attackers. Hence, The security concerns and protective measures of CL-based POI recommenders still require investigation.

### 5.3 Attacks Against Collaborative Learning Frameworks for Recommendations

Recently, various attacks have been proposed to threaten the security and privacy of collaborative learning frameworks for recommendations. These attacks include poisoning attacks [8, 24], inference attacks [8, 19, 24, 36], reconstruction attacks [7, 11], and byzantine attacks [29, 30]. In this work, we primarily focus on inference attacks. Zhang et al. [37] performed an investigation into privacy preservation for federated recommendations while their

study only exposed risks related to attribute-level information. In addition, Yuan et al. [36] designed an interaction-level inference attack to identify the set of interacted items by analyzing the user's uploaded parameters. However, focusing on traditional e-commerce recommendations, this method only considers the change of parameters about the target item, which is inefficient in POI recommendations. Instead, the proposed PITA identifies the set of visited POIs by combining the explicit information of the target POI and implicit information within its related POIs.

## 6 CONCLUSION

In this paper, we first conduct an in-depth analysis of privacy risks within CL-based POI recommenders and then design a novel attack named PITA to reveal users' historical trajectories in this scenario. Specifically, we identify the set of interacted POIs by combining explicit information from target POIs and implicit information within related POIs. We validate the efficacy of PITA on two datasets across two types of decentralized CL-based POI recommenders and empirical evidence unequivocally demonstrates the substantial threat posed by PITA to users' historical trajectories. Moreover, the conventional privacy-preserving method, LDP, has been proven ineffective in countering PITA. In light of this, we propose a novel defense mechanism based on an adversarial game called AGD to protect user privacy against PITA. It can accurately and effectively erase all sensitive POIs and their implicit information from related POIs with minimal impact on overall recommendation performance.

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
