# OpenReview forum: "Physical Trajectory Inference Attack and Defense in Decentralized POI Recommendation"
_ACM.org/TheWebConf/2024/Conference — TheWebConf24 Oral_

### Official Review · Reviewer_oXk7 · 2023-11-21

**Novelty:** 6
**Technical Quality:** 6

**Review:**

To address the privacy concerns related to users’ real mobility trajectories in decentralized collaborative learning POI recommendations, this paper introduces a new attack called Physical Trajectory Inference Attack (PITA) that aims to expose users’ historical trajectories by analyzing aggregated information from target POIs and their correlated POIs. The effectiveness of PITA is evaluated on two real-world datasets, and it is shown to pose a significant threat to users’ historical trajectories. Additionally, a defense mechanism called Adversarial Game Defense (AGD) is also proposed to eliminate sensitive POIs and their information in corrleted POIs. Overall, this paper offers a comprehensive analysis of the privacy risks in decentralized collaborative filtering recommender systems and proposes effective attack and defense mechanisms.
Pros:
1.	The motivation of this paper, i.e., addressing the privacy concerns  related to users’ real mobility trajectories in decentralized collaborative learning-based POI recommendations, is interesting and less explored.
2.	The proposed Physical trajectory inference attack (PTIA) and Adversarial game-based defense mechanism (AGD) make sense to me, and the finding that Local Differential Privacy (LDP) is proven ineffective against PITA is intriguing and meaningful.
3.	The paper conducts comprehensive experiments to evaluate the performance of the physical trajectory inference attack (PTIA) and the effectiveness of the corresponding defense mechanism based on an adversarial game (AGD).
Cons:
1.	At the end of Section 3.3, given differences between POI embeddings and initial distribution, it is not clear how to utlize elbow method to identify visited regions.
2.	The quality of figure 4 could be improved. Normally, the y-axis should be labelled with the metric it’s representing, such as F1-socre or hit ratio. However,  the second and third subgraphs label their y-axes with “DCLR” and “MAC”, which are identifiers for models rather than metrics, potentially leading to confusion. To clarify, the y-axis labels should state the name of the metric, and the legend should differentiate between the "DCLR" and "MAC" series to avoid any ambiguity for the reader.

**Questions:**

1.	How to utlize elbow method to identify visited regions?
2.	Why has the author not included the hit ratio for varying numbers of shadow sequences, denoted as 'V', in the Figure 4?

**Ethics Review Description:**

NULL

**Reviewer Confidence:**

4: The reviewer is certain that the evaluation is correct and very familiar with the relevant literature

**Scope:**

4: The work is relevant to the Web and to the track, and is of broad interest to the community

---

### Official Review · Reviewer_gF8J · 2023-11-21

**Novelty:** 6
**Technical Quality:** 6

**Review:**

This paper presents a study on inference attacks to the physical trajectory in POI recommendation tasks, which then motivates the design of the corresponding defensive mechanisms. The attack approach is able to tackle two different recommendation paradigms, i.e., model-sharing and knowledge distillation paradigms. The proposed defensive mechanism uses adversarial training to ensure that the sharable knowledge from either POI recommendation paradigms does not reveal users’ historical POI interactions. Overall, this work presents an interesting idea, and the developed methods are relatively easy to follow. My detailed comments on the pros and cons of this paper can be found below.

Pros:
+ Due to the location-sensitive nature of POI recommendation, it is of interest to discuss the level of privacy in state-of-the-art, decentralized POI recommender systems.
+ This paper considers two practical settings of the collaborative learning POI recommendation schemes, namely the model-sharing and the knowledge distillation schemes.
+ The proposed approaches are well-motivated and interesting, and their efficacy in attack and defense are respectively supported by experimental results.

Cons:
- Some more detailed background about the two collaborative learning paradigms is expected in the early parts of the paper (e.g., introduction and task definition), given that this notion is fairly new in POI recommendation.
- Some parts of the paper’s presentation can be enhanced. This applies to both textual and visual presentations. For example, in Section 4.1, “the effectiveness of PTIA” (should be PITA). Also in this section, the sentence “The situation varies when evaluating the performance of the defense mechanism” is not easily understandable and should be further clarified. In Figure 1, it is unclear how user’s POI model generates adversary’s knowledge logits.
- The experimental results can use a more extensive analysis. For example, the relationship between the number of interactions and the attack F1 after applying AGD can also be visualized (similar to Figure 3). Also, the changes in HR@10 when varying V can be plotted as well since the performance of the attacker PITA will likely to have an impact on the recommender during adversarial training.

**Questions:**

Q1. What is the connection between the POI inference attack in POI recommendation tasks and the membership inference attack in general recommendation tasks? Can those defensive mechanisms against membership inference attack be used for preventing POI inference attacks?

Q2. Is PITA and AGD generalizable to other collaborative learning (e.g., federated learning) POI recommender systems?

Q3. What is the efficiency gain after applying the region-based filtering strategy during the POI inference attack?

**Reviewer Confidence:**

4: The reviewer is certain that the evaluation is correct and very familiar with the relevant literature

**Scope:**

4: The work is relevant to the Web and to the track, and is of broad interest to the community

---

### Official Review · Reviewer_EkrW · 2023-11-23

**Novelty:** 5
**Technical Quality:** 4

**Review:**

This study introduces the Physical Trajectory Inference Attack (PITA) to highlight privacy risks in POI recommendation systems and proposes a novel defense mechanism (AGD) for enhanced security.

Pros:

Focused on Privacy: Addresses critical user privacy issues in LBSNs.

Innovative Methods: PITA exposes vulnerabilities; AGD counters these effectively.

Empirically Tested: Validated on real-world datasets, ensuring practical relevance.

Cons:

Implementation Complexity: AGD may be complex to integrate.

Specific Attack Focus: Solutions may be limited to countering PITA.

Privacy-Functionality Balance: Balancing enhanced privacy with recommendation efficiency could be challenging.

Overall, the study contributes significantly to privacy in LBSNs, offering a novel attack model and an effective defense mechanism, with a focus on maintaining recommendation performance.

**Questions:**

1. I recommend adding a diagram to illustrate the distinctions between model-sharing Collaborative Learning (CL) and Knowledge-Distillation CL for enhanced explanation.

2. Based on Figure 2, I think the public POI model, presumably STAN, which can learn visited POI features. So how do you get unvisited POI features?

3. Your task should be a click problem, but STAN is a sequence POI recommendation model. I think they are different tasks. Why do you choose STAN?

4. The experimental part should add the performance of STAN under the two datasets, that is, it does not include any attack and defense measures.

5. Why can the F1 metric evaluate the effectiveness of an attack?

6. While HR@10 might be an appropriate metric for click problems, POI recommendation tasks typically employ NDCG and Recall for a more holistic evaluation. Do you believe that the current metrics deployed in your study sufficiently capture the overall effectiveness of the model in this context?

**Reviewer Confidence:**

2: The reviewer is willing to defend the evaluation, but it is likely that the reviewer did not understand parts of the paper

**Scope:**

3: The work is somewhat relevant to the Web and to the track, and is of narrow interest to a sub-community

---

### Decision · Program_Chairs · 2024-01-22

**Decision:**

Accept (Oral)

**Comment:**

By summarizing the review comments and responses, this paper proposed a novel solution to solve the POI recommendation problem, and the experiment is solid and gets the sota results. However, the reviewers still have some concerns about the details of this paper, such as more background information, adding a diagram, etc. I recommend that the authors should fix these problems in their camera-ready version.